# *Azospirillum brasilense* and Zinc Rates Effect on Fungal Root Colonization and Yield of Wheat-Maize in Tropical Savannah Conditions

**DOI:** 10.3390/plants11223154

**Published:** 2022-11-18

**Authors:** Philippe Solano Toledo Silva, Ana Maria Rodrigues Cassiolato, Fernando Shintate Galindo, Arshad Jalal, Thiago Assis Rodrigues Nogueira, Carlos Eduardo da Silva Oliveira, Marcelo Carvalho Minhoto Teixeira Filho

**Affiliations:** 1Faculty of Agricultural and Veterinary Sciences, São Paulo State University (UNESP), Via de Acesso Prof. Paulo Donato Castellane s/n, Jaboticabal 14884-900, Brazil; 2Department of Plant Health, Rural Engineering, and Soils, São Paulo State University (UNESP), Av. Brasil, 56—Centro, Ilha Solteira 15385-000, Brazil; 3Center for Nuclear Energy in Agriculture, University of São Paulo (USP), Av. Centenário, 303—São Dimas, Piracicaba 13416-000, Brazil

**Keywords:** *Arbuscular mycorrhiza*, dark septate endophytes, *Zea mays* L., *Triticum aestivum* L., zinc fertilization, microbiomes

## Abstract

A successful microbial inoculant can increase root colonization and establish a positive interaction with native microorganisms to promote growth and productivity of cereal crops. Zinc (Zn) is an intensively reported deficient nutrient for maize and wheat production in Brazilian Cerrado. It can be sustainably managed by inoculation with plant growth-promoting bacteria and their symbiotic association with other microorganisms such as arbuscular mycorrhizal fungi (AMF) and dark septate endophytes (DSE). The objective of this study was to evaluate the effect of *Azospirillum brasilense* inoculation and residual Zn rates on root colonization and grain yield of maize and wheat in succession under the tropical conditions of Brazil. These experiments were conducted in a randomized block design with four replications and arranged in a 5 × 2 factorial scheme. The treatments consisted of five Zn rates (0, 2, 4, 6 and 8 kg ha^−1^) applied from zinc sulfate in maize and residual on wheat and without and with seed inoculation of *A. brasilense*. The results indicated that root colonization by AMF and DSE in maize–wheat cropping system were significantly increased with interaction of Zn rates and inoculation treatments. Inoculation with *A. brasilense* at residual Zn rates of 4 kg ha^−1^ increased root colonization by AMF under maize cultivation. Similarly, inoculation with *A. brasilense* at residual Zn rates of 2 and 4 kg ha^−1^ reduced root colonization by DSE under wheat in succession. The leaf chlorophyll index and leaf Zn concentration were increased with inoculation of the *A. brasilense* and residual Zn rates. The inoculation did not influence AMF spore production and CO_2_-C in both crops. The grain yield and yield components of maize–wheat were increased with the inoculation of *A. brasilense* under residual Zn rates of 3 to 4 kg ha^−1^ in tropical savannah conditions. Inoculation with *A. brasilense* under residual Zn rates up to 4 kg ha^−1^ promoted root colonization by AMF and DSE in the maize cropping season. While the inoculation with *A. brasilense* under 2 and 4 kg ha^−1^ residual Zn rates reduced root colonization by AMF and DSE in the wheat cropping season. Therefore, inoculation with *A. brasilense* in combination with Zn fertilization could consider a sustainable approach to increase the yield and performance of the maize–wheat cropping system in the tropical savannah conditions of Brazil.

## 1. Introduction

Brazil is the sixth-largest cereal producer in the world, producing 76% maize, 13% rice, 8% wheat and 1% sorghum [1]. Maize (*Zea mays* L.) is the second-largest produced cereal crop in Brazil, while wheat (*Triticum aestivum* L.) production is still not sufficient to satisfy national demands. Therefore, Brazil imports a huge amount of wheat from other countries to feed the nation [2]. The cereal production of tropical savannah may be restricted by several factors such as soil-borne diseases, soil erosion and low nutrient and water use efficiency; rhizosphere microbiome and soil-existing microorganisms [3,4]. Thus, cereals cultivation in tropical savannah could be treated as a matter of food security due to their short life cycle, contribution to carbon (C) sequestration and soil organic matter, which can improve the nutrient pool and use efficiency and crop productivity [5].

The intense weathering of tropical soils has devastating effects on soil organic matter and nutrients use efficiency, especially Zn, leading to low cereal productivity and soil fertility [6]. Zinc deficiency is one of the globally recognized micronutrient deficiencies that limit crop growth and productivity [7]. Zinc starvation affects the physiological and enzymatic activities and protein synthesis of soil and plants [8]. Zinc is one of the fundamental constituents in the synthesis of tryptophan, indole acetic acid (IAA), proteinases, peptidases, dehydrogenases and phosphohydrolases, which could improve soil and plant health [9]. The deficit or excess of Zn may have harsh consequences on plant health and yield; therefore, proper Zn management is a wise agronomic strategy to improve soil health and crop productivity [10,11].

The management of the interaction of soil–plant microorganisms is alternative strategy that could contribute to soil–plant health and productivity [12]. Plant growth-promoting bacteria (PGPBs) such *Azospirillum* has the ability to fight for colonization site in above and below soil parts of several cereal crops and improve synthesis of phytohormones [13,14,15]. The combined application of *Azospirillum brasilense* and Zn can increase production of several phytohormones that are involved in plant growth [16]. The beneficial activities of PGPBs can improve soil microbial diversity and root colonization by developing symbiotic association between plants and several beneficial fungi [17,18]. These symbiotic fungi may be obligated biotrophs, arbuscular mycorrhizal fungi (AMF) or facultative biotrophs, melanized septate endophytics (dark septate endophytic—DSE) [19].

Nitrogen-fixing bacteria, particularly *A. brasilense*, can effectively improve plant growth, nutrients uptake and colonization by association with microsymbionts in root rhizosphere [20]. *Azospirillum* could positively contribute mycorrhizal colonization due to its activity as mycorrhiza helper bacterium [17]. *A. brasilense* is involved in the promotion of fungal propagation and germination, stimulate mycelial growth or alter root architecture through the production of growth factors [18]. Inoculation with *A. brasilense* also contributed to nutrient mobilization that could increase mycorrhizal performance, root colonization and crop productivity [20]. This bacterium develops a positive correlation with native mycorrhizal fungi of rhizosphere that can stimulate and support root colonization of exogenous applied AMF, thus promoting plant growth, biomass and yield [20,21].

Arbuscular mycorrhizal fungi encourage plant growth and development by ensuring tolerance against biotic and abiotic stresses [22]. AMF colonize in the root rhizosphere through its hyphae and ensuring water and nutrients acquisition for the host plants. The effect of AMF on plant growth is more significant in regard of nutrients with low mobility in the soil, such as phosphorus (P) and Zn [23]. In addition, DSEs are ascomycetes that develop microsclerotia and support inter- and intracellular colonization within root rhizosphere of several plants [23,24]. DSEs are similar in functions to AMF, promoting the absorption of water and nutrients by their host plants [25]. However, dissimilar to AMF, some DSE fungi are able to synthesize auxins and other organic compounds that stimulate root growth. These AMF and DSE fungi are simultaneously occupying the same cortical region of the roots system, where these fungi can interact with other microorganisms, such as *A. brasilense* [26]. However, there are few studies on the influence of residual Zn rates and inoculation with *A. brasilense* on AMF and DSE colonization and productivity of maize and wheat crops in tropical savannah conditions.

Therefore, it is essential to adapt management and biotechnological strategies to minimize environmental damage and increase agricultural production in Brazilian Cerrado soils. The hypothesis of this research is that residual Zn rates in association with the inoculation of *A. brasilense* would increase root fungi colonization and AMF spores production, leading to higher maize and wheat productivity. Therefore, this study aimed to evaluate the effect of *A. brasilense* inoculation and residual Zn rates on root fungi colonization, spores production of AMF, chlorophyll and leaf Zn contents and grain yield of maize and wheat in succession under tropical savannah conditions.

## 2. Results

### 2.1. Treatments Effect on Maize

The AMF colonization in maize root was higher with inoculation of *A. brasilense* and 4 kg Zn ha^−1^ as compared to non-inoculated plants (Figure 1A). There was a nonlinear response of AMF root colonization to increasing Zn rates up to 3.5 kg Zn ha^−1^ in the presence of *A. brasilense*, whereas the Zn rates up to 4.4 kg Zn ha^−1^ in the absence of inoculation were observed with higher AMF colonization in the maize root rhizosphere (Figure 1A). In contrast, higher root DSE colonization was observed without the inoculation of *A. brasilense* in combination with the application of 2 and 6 kg Zn ha^−1^, while the treatments with inoculation of *A. brasilense* performed better with the application of 8 kg Zn ha^−1^ (Figure 1B). The root DSE colonization was linearly adjusted to increasing Zn rates in the presence of *A. brasilense* inoculation and nonlinearly up to 4.3 kg Zn ha^−1^ without inoculation treatments (Figure 1B). Number of AMF spores and released CO_2_-C were not affected by the Zn rates or inoculation (Figure 1C,F).

The interaction of inoculation with *A. brasilense* and Zn rates was not significant for the leaf chlorophyll index and leaf Zn concentration of maize (Figure 2). The single effect of inoculation with *A. brasilense* and Zn rates positively affected leaf chlorophyll index and leaf Zn concentration of maize. Inoculation with *A. brasilense* improved the leaf chlorophyll index and leaf Zn concentration of maize as compared to without inoculation (Figure 2A,C). In addition, the leaf chlorophyll index of maize was linearly increased with the increasing Zn rates (Figure 2B), while leaf Zn concentration was showed a nonlinear response with increasing Zn rates. The leaf Zn concentration of maize was increased with increasing Zn rate up to 3.9 kg ha^−1^, while a further increase in the Zn rates caused a decrease in the leaf Zn concentration of maize (Figure 2D).

The grain yield and yield components of maize were significantly influenced by Zn rates and *A. brasilense* inoculation (Figure 3). Inoculation with *A. brasilense* produced a higher number of grains cob^−1^ (Figure 3A), heavier hundred grains weight (Figure 3C) and greater grain yield (Figure 3E) of maize as compared with uninoculated treatments. Inoculation with *A. brasilense* increased the grain yield by 4.3% as compared to without inoculation treatments (Figure 3E).

In addition, number of grains cob^−1^ of maize were nonlinearly increased with the increasing Zn rates, where the number of grains cob^−1^ were increased with the increasing Zn rates up to 4.3 kg ha^−1^ (Figure 3B). Hundred-grain weight of maize was increased with increasing the Zn rate up to 4.2 kg ha^−1^, while a further increase in the Zn rates led to lighter grains (Figure 3D). The grain yield of maize showed a nonlinearly response with increasing the Zn rates (Figure 3F). A greater grain yield (8586 kg ha^−1^) was observed at the maximum applied Zn rates up to 3.8 kg ha^−1^, whereas a further increase in the Zn rates decreased the grain yield of maize (Figure 3F).

### 2.2. Treatment Effect on Wheat

The colonization of AMF in root rhizosphere of wheat was higher with the inoculation of *A. brasilense* either in the absence of residual Zn rates or with 2 and 4 kg Zn ha^−1^ residual rates as compared to the other treatments (Figure 4A). Root AMF colonization was linearly decreased with increasing the Zn rates under inoculation of *A. brasilense* (Figure 4A). In addition, root DSE colonization was linearly increased with increasing the residual Zn rates regardless of inoculation with *A. brasilense* (Figure 4B).

A number of AMF spores were linearly decreased with the increasing residual Zn rates (Figure 4C). It was also observed that AMF sporulation increased (38 × 100 g dry soil) with the inoculation of *A. brasilense*, an increase of 11.8% was observed as compared to without the inoculation treatments (34 × 100 g dry soil) (Figure 4D). Released CO_2_-C responded nonlinearly to increasing residual Zn rates up to 3.6 kg Zn ha^−1^ (Figure 4E). In addition, released CO_2_-C was increased by 6.3% with the inoculation of *A. brasilense* as compared to non-inoculated treatments (Figure 4F).

The interactions of the inoculation and residual Zn rates were not significant for the leaf chlorophyll index and leaf Zn concentration of wheat. In addition, the single effect of without or with inoculation of *A. brasilense* also did not influence the leaf chlorophyll index and leaf Zn concentration of wheat (Figure 5A,C). Although, the leaf chlorophyll index and leaf Zn concentration of wheat were adjusted to nonlinear regression with increasing residual the Zn rates up to 5.2 and 4.7 kg Zn ha^−1^, respectively (Figure 5B,D). Further increase in the Zn rates led to a reduced concentration of leaf chlorophyll index and leaf Zn concentration of wheat (Figure 5B,D).

Grain yield and yield components of wheat were positively influenced by single treatment effect of inoculation and residual Zn rates (Figure 6). Inoculation with *A. brasilense* significantly influenced the grains spike^−1^ and 100-grain weight of wheat as compared to without inoculation (Figure 6A,C).

The grain yield of wheat was not affected by the inoculation with *A. brasilense* (Figure 6E). Grains spike^−1^, 100-grain weight and grain yield of wheat were adjusted to be nonlinear with the increasing Zn rates (Figure 6B,D,F). Grains spike^−1^ were increased with the maximum increasing residual Zn rates up to 4 kg ha^−1^, while a further increase in the Zn rates decreased the number of grains spike^−1^ of wheat (Figure 6B). Heavier 100 grains were observed at the maximum Zn rate of 4.1 kg ha^−1^, while a further increase in the Zn residual rate lowered the 100-grain weight of wheat (Figure 6D). The grain yield of wheat was nonlinearly increased with the increasing Zn rates up to 4.7 kg ha^−1^ (Figure 6F).

## 3. Discussion

The present results indicated that root AMF colonization was fluctuated with residual zinc (Zn) rates. The root AMF colonization in maize and wheat was increased with 2 to 4 kg ha^−1^ of residual Zn along with inoculation of *A. brasilense* (Figure 1A and Figure 4A). It might be possible due to positive association of beneficial microbes, where they develop root exudates that trigger fungus growth and colonization in root rhizosphere [21]. The interaction of rhizobacterial populations and mycorrhizal fungi encourage plant growth by alleviating stress condition and modulating rhizosphere microbiome [27]. In addition, Zinc regulates root system of crop plants by contributing to several phytohormones that can indirectly stimulate root colonization of AMF in rhizosphere of host plants [28,29].

Bidondo et al. [30] indicated that bacteria can interact with the native microbial community in mycorrhizosphere inside spores or mycelia. It has verified in the present study that AMF spores production were increased with inoculation of *A. brasilense* and residual Zn rates under successive wheat cultivation (Figure 1D,E). It also is possible due to the role of diazotrophic bacteria in promotion of AMF colonization and spores number, which improve rhizospheric conditions and lad to a better performance and quality of plants [31]. Several bacterial strains of *Azospirillum* spp. and *Pseudomonas* spp. along with Zn application increased root development and formation of new infection points that increase AMF spores production and colonization [32,33], as proved in the present study with inoculation of *A. brasilense* in maize and wheat cultivation (Figure 1D and Figure 4D; Appendix A).

The colonization of DSE in root system of wheat or maize and its interaction with other endophytic microorganisms is still a point of discussion and need to be addressed. The current results indicated that root colonization of DSE in maize and wheat cropping system was increased with increasing Zn rates (Figure 1A,B and Figure 4A,B). The might be due to the ability of DSE to increase tolerance against hyper-accumulation of metals [34]. The simultaneously colonization of DSE with AMF in plant root system under optimal Zn doses counter the harsh effects of biotic and abiotic stresses [35]. The current results verified that low Zn rates showed a non-mutualistic relationship between host plants and DSE fungus, which can reduce root biomass production regardless of inoculation. In this scenario, higher Zn rates increased root colonization by DSE as verified in the present study (Figure 1B and Figure 4B). Liu et al. [36] reported that DSE fungus has the ability to increase colonization under higher Zn doses by increasing Zn compartment in roots cell wall that can reduce Zn proportion in root soluble fraction and cell organelles.

*Azospirillum brasilense* has the ability to produce siderophores and other molecules like salicylic acid that may decrease mycelial growth [20]. The present study also showed a reduction in DSE colonization in roots of wheat (Figure 1B and Figure 4B). This might be due to bacterial role in the inhibition of mycelia growth and fungal colonization. In contrast, there has not been reported any antagonistic effect between *A. brasilense* and DSE in root/soil system [37]. Despite this, Santos et al. [38] reported a mutualistic association between these microorganisms in most situations, however, they can also develop a pathogenic affect to each other. DSEs colonize in the roots of host plants through mineralization and release of nutrients into the soil solution that are absorbed by the plants [39]. Plants can choose one of the fungi (AMF or DSE) on the basis of favorable environmental conditions, plant needs and energy expenditure for their survival [40]. Root colonization can be increased by inoculation with *A. brasilense*, which increases roots volume points of infection by AMF or DSE [39,41]. However, the interaction is still unclear that needs further studies on the influence of molecules produced by *A. brasilense* in root DSE colonization and their interaction with AMF.

The CO_2_-C released in maize root did not show significant effect with inoculation or Zn fertilization (Figure 1E,F). While the inoculation with *A. brasilense* under residual Zn rates increased CO_2_-C activity in wheat root system (Figure 4E,F). The reason might be due to the influence of crop and soil management under microbial activities. The respiratory activities of soil microorganisms (bacteria, fungi and other) are responsible for the release of CO_2_-C and being act as sensitive indicators for soil quality [42]. In addition, different crop practices or even external inoculation of microorganism can generate changes in soil microbial activities. Bera et al. [43] observed an increasing trend in microbial respiratory activities under wheat succession to rice in a no-tillage system and reported that respiratory activities were increased from sowing to flowering while decreasing in later maturity. The present results reflected similar behavior for wheat succession to maize at 110 days after emergence. It was also reported that there were no significant differences in the CO_2_-C release at wheat maturity in succession to maize during 24-h observation [44,45]. These results corroborate the low values of CO_2_-C in the present study (Figure 1F and Figure 4F), which could lead to stabilized environment where a higher carbon as microorganism biomass is incorporated into soil and a low value of CO_2_-C is released lost to atmosphere.

Our results indicated that residual Zn rates and inoculation with *A. brasilense* increased chlorophyll index, leaf Zn concentration of maize and wheat (Figure 2 and Figure 5). It is possible that *A. brasilense* has important role in several metabolic and critical plant processes including nutrient acquisition, biological nitrogen fixation and hormones production [13,46,47], promote growth of maize–wheat cropping system in tropical savannah [48]. It has been reported that Zn fertilization along with bacterial inoculation is a sustainable management strategy that increase plant performance, nutrient uptake and nutrient use efficiency in cereal cropping system [6,49]. The current results were further supported by the findings that inoculation with *A. brasilense* in combination with soil Zn application increased growth traits and leaf Zn nutrition of wheat and maize [6,39,40]. In addition, it has also been reported that *A. brasilense* inoculation increase leaf nutrient concentration and chlorophyll index of cereal crops [50,51].

The grain yield and yield components of maize and wheat were significantly different with Zn fertilization and inoculation (Figure 3 and Figure 6). The grain yield of both the crops showed a nonlinear regression adjustment (Figure 3F and Figure 6F). This might be due to the medium Zn soil Zn content in the initial soil test (Table 1), which could meet the needs of plants and also explain the decreasing trend in productivity with further increase in Zn rates (Figure 3F and Figure 6F). In addition, the inoculation with *A. brasilense* showed an increase in grain production (Figure 3F and Figure 6F). It is possible that *A. brasilense* favored the development of root system with higher absorption of nutrients and water that has a positive influence on the nutritional status of the plant [6,52]. The assimilation of water and nutrients to spike and shoot are directly related to plant nutritional status [50], thus, leading to higher grain productivity. Our results showed that a grain yield of maize and wheat was increased with the inoculation of *A. brasilense* as compared to without inoculation (Figure 3E and Figure 6E). The average wheat production (1637 kg ha^−1^) with inoculation of *A. brasilense* in the current study is higher than average wheat production (900 kg ha^−1^) of State of Mato Grosso do Sul [53].

## 4. Materials and Methods

### 4.1. Site Description

The study was conducted under field conditions in Selvíria (Brazilian Cerrado region), State of Mato Grosso do Sul, Brazil (20°22′ S and 51°22′ W, 335 m above sea level (Figure 7) during 2013/14 (maize) and 2014 (wheat).

The soil was classified as Rhodic Haplustox (clayey Oxisol), according to the Soil Survey Staff [54]. Twenty random soil samples were collected from the entire experimental site with a soil core type cup auger (0.10 m × 0.40 m—cup diameter and length, respectively) from a 0.00–0.20-m depth before the initiation of the field trial. The soil samples collected from each depth were homogenized, air-dried, sieved (2 mm) and stored at room temperature. The soil chemical attributes [55] and granulometry characterization [56] are summarized in Table 1. In addition, twenty soil samples with root rhizospheres were also randomly collected before planting of the respective crops to characterize colonization by *Azospirillum* sp. in the experimental area. The samples were mixed to obtain a random composite sample. The fresh sample was analyzed by the most probably number (MPN) technique with serial dilutions and inoculations in vials containing semisolid NFb medium without N addition, followed by growth at 35 °C for 48. The MPN of *Azospirillum* sp. was recorded using the McCrady table with three repetitions per dilution [57].

The experimental area had been cultivated with annual leguminous and cereal crops for over 28 years. In addition, the area has been under no-tillage cultivation system for the last 13 years. The crop sequence prior to field trial was fallow until 2013 and black oats (*Avena strigosa* Schreb.) in 2013. Maximum, average and minimum temperatures and rainfall observed during the field trial are presented in Appendix A.

### 4.2. Experimental Design and Treatments

The experimental was designed in a randomized complete block with four replicates, arranged in a 5 × 2 factorial scheme. The treatments were consisted of five Zn rates (0, 2, 4, 6 and 8 kg Zn ha^−1^) applied from zinc sulphate (20% Zn and 10% S) and two seed inoculation with *A. brasilense* (without or with). The total area of each experimental plot was 13.5 m^2^, comprised of six maize rows of five meters long at row space of 0.45 m. Wheat were cultivated in twelve rows of five meters and 0.17 m apart with a plot total size of 10.2 m^2^. The useful area of maize–wheat plot were central rows (10 m^2^).

The seeds of maize and wheat were treated with insecticide and fungicide before inoculation. The seeds of both crops were treated with fungicides (carbendazim + thiram at an active ingredient (a.i.) of 45 g + 105 g per 100 kg seed) and insecticides (imidacloprid + thiodicarb at (a.i.) of 45 g + 135 g per 100 kg seed) before inoculation. This is a general practice used by cereals growing farmers however, the influence of chemical seed treatment on inoculation efficiency of PGPB is still controversial [58,59,60]. The inoculation with *A. brasilense* of maize or wheat seeds was carried out by mixing and coating inoculant manually in plastic bags before an hour of plantation.

The *A. brasilense* strains Ab-V5 and Ab-V6 (CNPSo 2083 and CNPSo 2084 respectively, guarantee of 2 × 10^8^ colony forming unity (CFU) mL^−1^) were applied to maize seeds (24 kg) at a rate of 200 mL liquid inoculant ha^−1^ and 300 mL ha^−1^ to 150 kg of wheat seeds. These strains under similar conditions (specifically Brazilian Cerrado) showed positive results on maize and wheat development [47,50,51]. The draft genome sequences of *A. brasilense* strains Ab-V5 and Ab-V6 carry similar *fix* and *nif* genes which are linked to biological N fixation [13]. These strains have different hormones production capacity however, sharing the same gene for auxin production [61]. In addition, Ab-V5 and Ab-V6 have the capacity to induce expression of genes associated to abiotic and biotic stress tolerance in plants [62]

Zinc rates (0, 2, 4, 6 and 8 kg ha^−1^) were manually applied to soil surface at even distribution in maize crop. The calculated amount of fertilizer (zinc sulphate) per plot was applied in between rows at V_2_ stage of maize (with two leaves completely unfolded). The experimental area was irrigated with central pivot irrigation system (14 mm) soon after side-dress Zn application to incorporate fertilizer in soil. The Zn fertilizer was not applied in wheat crop in order to analyze the residual effect of the treatments.

### 4.3. Field Management

#### 4.3.1. Maize

The area was broadcast applied with limestone (composed of 28% CaO and 20% MgO with an effective neutralizing power of 88%) at the rate of 2.5 Mg ha^−1^, 65 days before maize sowing. The amount of lime applied was based on initial soil analysis and base saturation to 70%, following Equation (1).
(1)$$\mathrm{LN}=\frac{\mathrm{CEC}\;\left(V2-V1\right)}{10\times \mathrm{ENP}}$$
where LN = Limestone required in Mg ha^−1^, CEC = cation exchange capacity, V2 = bases saturation to be achieved, V1 = current based saturation and ENP = effective neutralization power.

Pre-experiment weeds were controlled by spraying 2, 4-D (670 g ha^−1^ a.i.) and glyphosate (1800 g ha^−1^ a.i.). A maize triple hybrid DKB 350 VT PRO was sown on 4 December 2013 by placing 3.3 seeds per meter. The NPK 08-28-16 (32, 112 and 64 kg ha^−1^ of N, P_2_O_5_ and K_2_O respectively) was applied at a dose 400 kg ha^−1^ in plantation, based on soil analysis. The crop was irrigated with a center pivot irrigation system at 14 mm water supply. Seeds were emerged five days after sowing in both growing seasons. The recommended N (150 kg N ha^−1^ as ammonium sulfate, which contains 21% N and 23% S) was applied manually in side-dress on V6 growth stage. The post emergence weeds during crop development were controlled by the application of atrazine (1000 g ha^−1^ a.i.) and tembotrione (84 g ha^−1^ a.i.), along with vegetable oil adjuvant (720 g ha^−1^ a.i.). In addition, triflumuron (24 g ha^−1^ a.i.) and methomyl (215 g ha^−1^ a.i.) were used for controlling insects. The plants were harvested manually at 108 DAE (27 March 2014).

#### 4.3.2. Wheat

Wheat planting was carried out on exact area of preceding crop (maize) to analyze residual effect of Zn applied treatments in wheat (successor crop). The wheat genotype CD 116 was sown (80 seeds per meter) on 16 May 2014. A basal fertilization of 350 kg ha^−1^ of N-P-K (08-28-16) applied at plantation was equivalent to 28, 98 and 56 kg ha^−1^ of N, P_2_O_5_ and K_2_O. Seedling were emerged five days after sowing. The area was irrigated with a center pivot irrigation system adjusted to a 14-mm water depth at the intervals of 72 h approximately. Nitrogen (120 kg N ha^−1^ as ammonium sulfate, which contains 21% N and 23% S) was manually applied at a growth stage GS21 [63] in an even distribution of fertilizer to all treatments on soil surface. The post-emergence weeds were controlled by metsulfuron-methyl (3 g ha^−1^ a.i.). The crop was manually harvested at 110 DAE (9 September 2014).

### 4.4. Measurements

The microbiological evaluations were performed by collecting four soil samples with maize or wheat roots at the depth of 0.00–0.10 m in each experimental plot. The samples were collected at flowering stage of maize and wheat. The collected roots were washed and stored in a 50% alcohol solution. One gram of root per plot was clarified in KOH 10% and HCl 1% solution, stained with trypan blue 0.05% and stored in lactoglycerol to assess root AMF and DSE colonization [64]. Root colonization was determined by evaluating 100 segments of fine roots per plot.

The soil samples were homogenized and respiratory activity were determined by quantifying carbon released as CO_2_-C in 100 g fresh soil per plot, following the methodology of Anderson and Domsch [44].

The remaining of collected soil was air-dried, sieved (2 mm) and stored at room temperature. The number of AMF spores were determined from 100 g dry soil sample per plot. The AMF spores were separated from the soil according to methods of decantation and wet sieving [65], centrifugation and sucrose flotation [66]. Acrylic plate with concentric rings were used to count the spores under a stereoscopic microscope (40×).

The leaf chlorophyll index (LCI) was measured at flowering stage of maize and wheat in each plot using a portable chlorophyll Falker meter (ClorofiLOG^®^—model CFL—1030 Falker, Porto Alegre, Brazil). Twenty leaves of maize and 30 of wheat were collected at flowering stage of each crop in labeled paper bags, dried in an air tight oven and quantified the leaf Zn concentration with the methodology of Malavolta et al. [67].

Grains spike^−1^/cob^−1^ were calculated from the ten representative spikes and cob at the harvest of wheat and maize respectively. Hundred-grains weight was quantified at harvest through a digital scale. Grain yield was determined by spikes collection in useful lines of each maize and wheat plots. The grains were quantified after mechanical threshing and the data processed in kg ha^−1^ to 13% (humidity).

### 4.5. Statistical Analysis

All data were initially tested for normality using Shapiro and Wilk [68] test and Levene’s homoscedasticity test (*p* < 0.05) which showed the data to be normally distributed (W ≥ 0.90). The data was then analyzed by ANOVA (F test) in a 2-way factorial design with Zn application rates and *A. brasilense* inoculation, their interaction was considered fixed effects in the model while block was considered a random variable. Mean separation was done for significant of main or interaction effects using Tukey test. Regression analysis was also performed to assess whether there is a linear or non-linear response to Zn rates using R software [69].

## 5. Conclusions

Root colonization by AMF and DSE in maize–wheat cropping system were positively increased with seed inoculation of *A. brasilense* under residual Zn rates. Inoculation with *A. brasilense* at residual Zn rates of 4 kg ha^−1^ increased root colonization by AMF under maize cultivation. Inoculation with *A. brasilense* at residual Zn rates of 2 and 4 kg ha^−1^ reduced root colonization by DSE under wheat in succession. Number of spores of AMF in maize root system was not significant while these spores and released carbon (CO_2_-C) were increased with inoculation of *A. brasilense.* Leaf chlorophyll index and leaf Zn concentration were also improved with single effect of inoculation and residual Zn rates. It was also concluded from our results that grain yield and yield components of maize–wheat increased with inoculation under residual Zn rates of 3 to 4 kg ha^−1^ in tropical savannah conditions.

## Figures and Tables

**Figure 1 plants-11-03154-f001:**
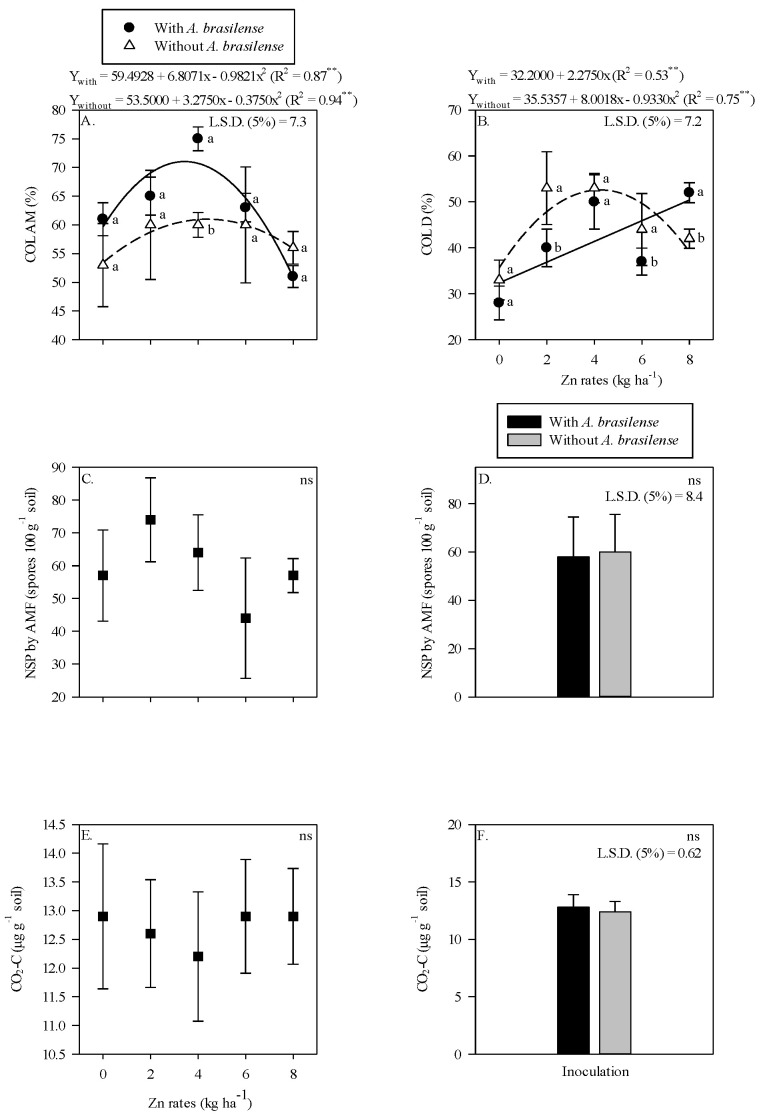
Root colonization of arbuscular mycorrhizal—COL AM (**A**), dark septate—COL D (**B**) as a function of Zn rates and *Azospirillum brasilense* interaction while number of spores (NSP) of arbuscular mycorrhizal fungi and released carbon from CO_2_ (CO_2_-C) in maize roots system as a function of residual Zn rates (**C**,**E**) and inoculation or not with *Azospirillum brasilense* (**D**,**F**). The letters correspond to a significant difference at 5% probability level (*p* ≤ 0.05). ** and ns = significant at *p* < 0.01 and not significant, respectively. Error bars indicate standard deviation of means (*n* = 4). L.S.D. (5%) = Least significant difference.

**Figure 2 plants-11-03154-f002:**
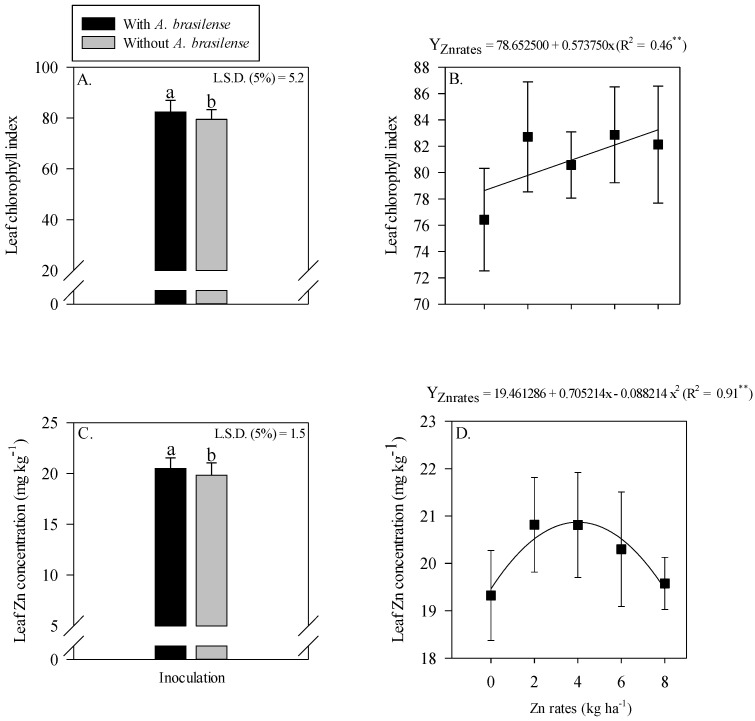
Leaf chlorophyll index and leaf zinc (Zn) concentration of maize as a function of single effect of inoculation or not with *Azospirillum brasilense* (**A**,**C**) and Zn rates (**B**,**D**). The letters correspond to a significant difference at the 5% probability level (*p* ≤ 0.05). ** = significant at *p* < 0.01. Error bars indicate standard deviation of means (*n* = 4). L.S.D. (5%) = Least significant difference.

**Figure 3 plants-11-03154-f003:**
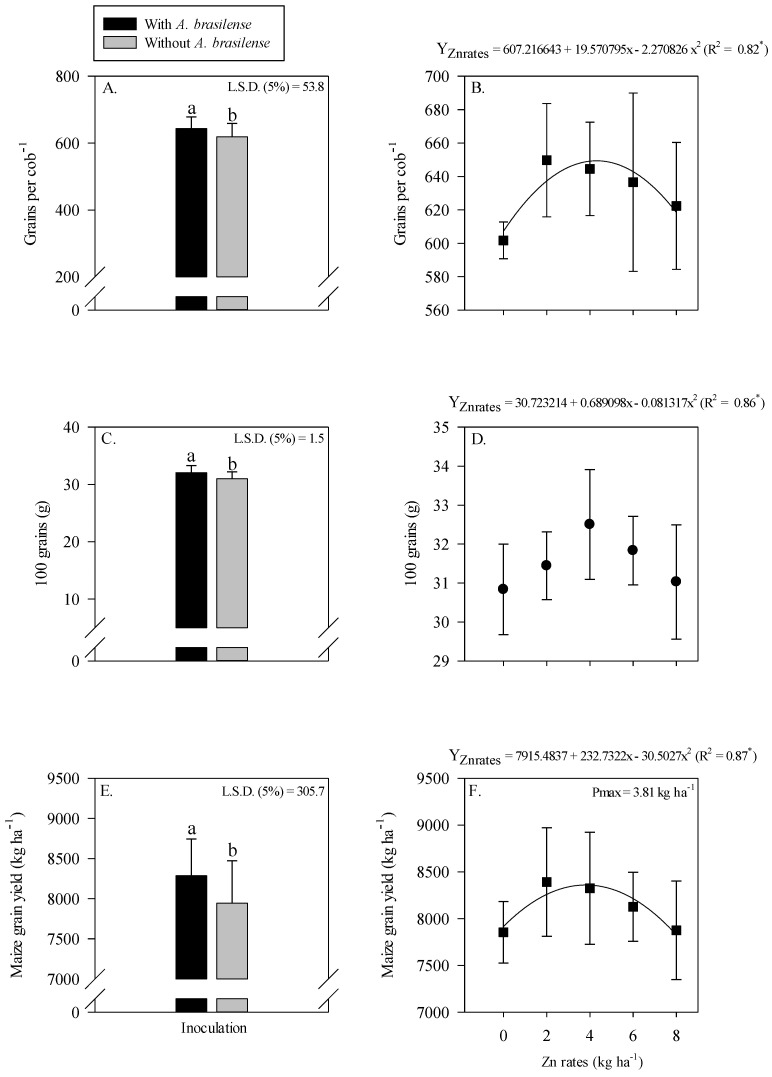
Number of grains cob^−1^, 100-grains weight and grain yield of maize as a function of single effect of inoculation or not with *Azospirillum brasilense* (**A**,**C**,**E**) and Zn rates (**B**,**D**,**F**). The letters correspond to a significant difference at 5% probability level (*p* ≤ 0.05). * = significant at *p* < 0.05. Error bars indicate the standard deviation of the mean (*n* = 4). L.S.D. (5%) = Least significant difference.

**Figure 4 plants-11-03154-f004:**
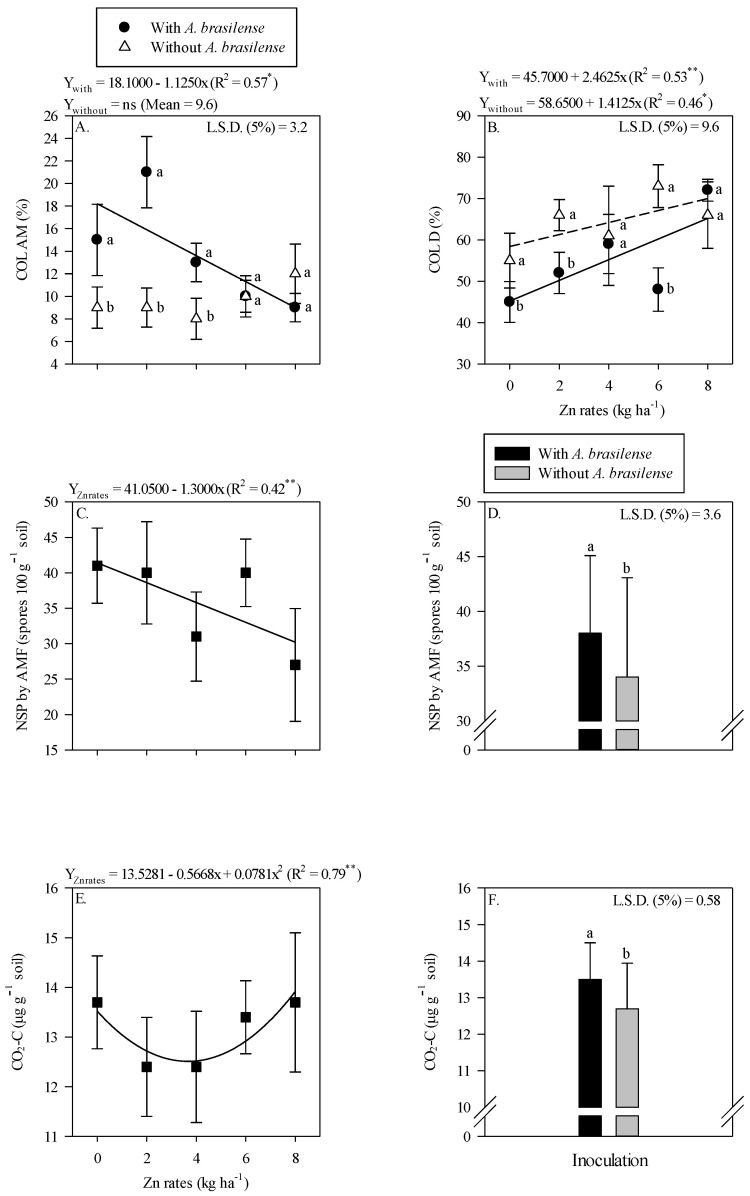
Root colonization of arbuscular mycorrhizal—COL AM (**A**), dark septate—COL D (**B**) as a function of the interaction of Zn rates and *Azospirillum brasilense* inoculation, while a number of spores (NSP) of arbuscular mycorrhizal fungi released carbon from CO_2_ (CO_2_-C) in the wheat root system as a function of the single effect of the Zn rates (**C**,**E**) and inoculation or not with *A. brasilense* (**D**,**F**). The letters correspond to a significant difference at the 5% probability level (*p* ≤ 0.05). ** and * significant at *p* < 0.01 and significant at *p* < 0.05, respectively. Error bars indicate standard deviation of means (*n* = 4). L.S.D. (5%) = Least significant difference.

**Figure 5 plants-11-03154-f005:**
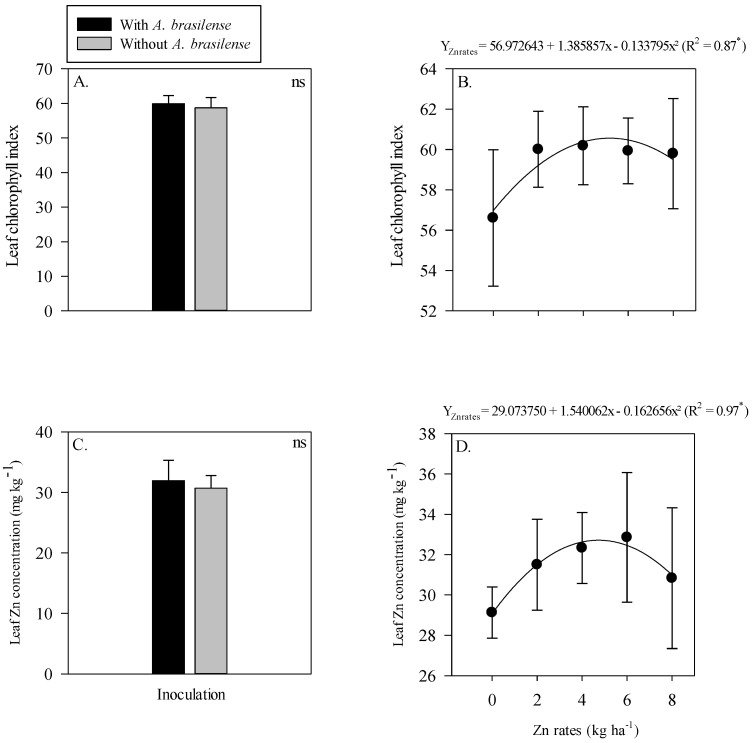
Leaf chlorophyll index and leaf zinc (Zn) concentration of wheat as a function of single effect of inoculation or not with *Azospirillum brasilense* (**A**,**C**) and the Zn rates (**B**,**D**). The letters correspond to a significant difference at the 5% probability level (*p* ≤ 0.05). * and ns = significant at *p* < 0.05 and not significant, respectively. Error bars indicate standard deviation of means (*n* = 4). L.S.D. (5%) = Least significant difference.

**Figure 6 plants-11-03154-f006:**
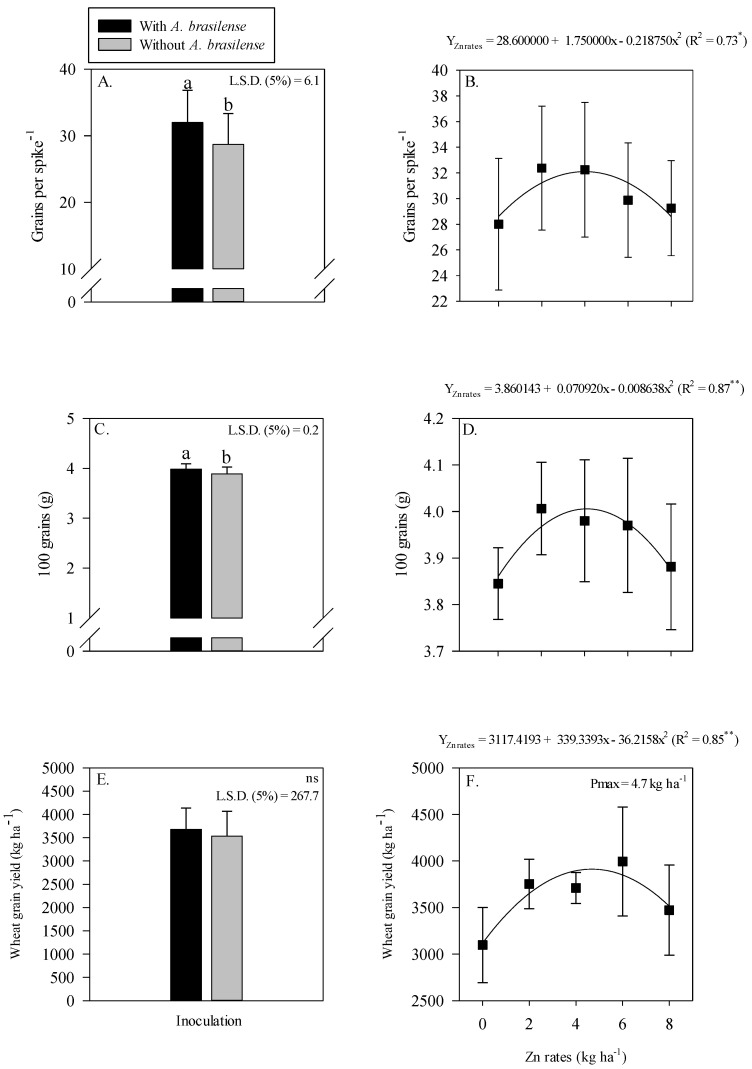
Number of grains spike^−1^, 100-grain weight and grain yield of wheat as a function of single effect of inoculation or not with *Azospirillum brasilense* (**A**,**C**,**E**) and the Zn rates (**B**,**D**,**F**). The letters correspond to a significant difference at the 5% probability level (*p* ≤ 0.05). **, * and ns = significant at *p* < 0.01, *p* < 0.05 and not significant, respectively. Error bars indicate the standard deviation of the mean (*n* = 4). L.S.D. (5%) = Least significant difference.

**Figure 7 plants-11-03154-f007:**
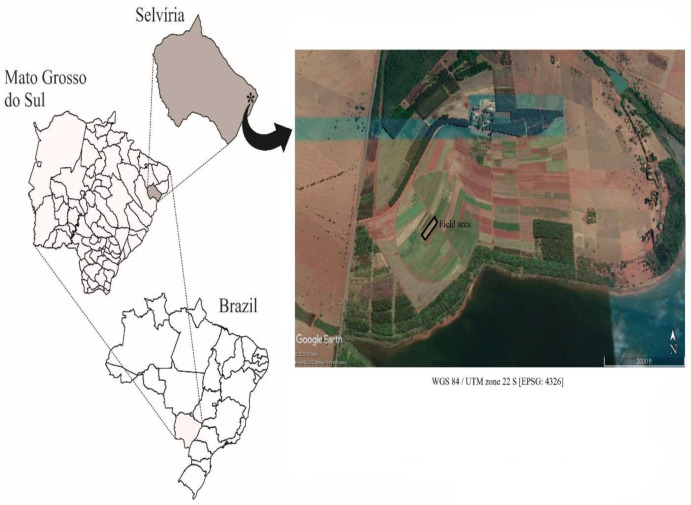
Study area at Selvíria, State of Mato Grosso do Sul, Brazil (20°22′ S, 51°22′ W, the altitude of 335 m above sea level). Map created by using QGIS software and Google Earth program. QGIS Development Team (2019). QGIS Geographic Information System. Open Source Geospatial Foundation Project. http://qgis.osgeo.org (accessed on 10 January 2022). Image obtained in Google Earth program. Google company (2020).

**Table 1 plants-11-03154-t001:** Soil physiochemical attributes and *Azospirillum* sp. populations in the 0–0.20-m layer before field trial beginning.

Soil Chemical Attributes	0–0.20-m Layer
P (resin)	13.0 mg dm^−3^
S (SO_4_)	6.0 mg dm^−3^
Organic matter	23.0 g dm^−3^
pH (CaCl_2_)	4.8
K	2.6 mmol_c_ dm^−3^
Ca	13.0 mmol_c_ dm^−3^
Mg	8.0 mmol_c_ dm^−3^
H + Al	42.0 mmol_c_ dm^−3^
Al	2.0 mmol_c_ dm^−3^
B (hot water)	0.24 mg dm^−3^
Cu (DTPA)	5.9 mg dm^−3^
Fe (DTPA)	30.0 mg dm^−3^
Mn (DTPA)	93.5 mg dm^−3^
Zn (DTPA)	0.5 mg dm^−3^
Cation exchange capacity (pH 7.0)	65.6 mmol_c_ dm^−3^
Base saturation (%)	36
Sand	438 g kg^−1^
Silt	90 g kg^−1^
Clay	472 g kg^−1^
*Azospirillum* sp. most probably number	1.65 × 10^4^ CFU g^−1^ soil

*n* = 20, DTPA =diethylenetriaminepentaacetic acid.

## Data Availability

The datasets generated during and/or analyzed during the current study are available from the corresponding author on reasonable request.

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
