# Peer review of "Azospirillum brasilense and Zinc Rates Effect on Fungal Root Colonization and Yield of Wheat-Maize in Tropical Savannah Conditions"

_plants, 2022, doi:10.3390/plants11223154_

Round 1
Reviewer 1 Report (Previous Reviewer 2)
The authors did a great effort addressing my previous comments and, I guess, the comments from other reviewers to improve the manuscript. Data are now better presented and text is more complete. However, there are still some issues remained.
1. Discussion is still rather shallow. It should be improved by highlighting novel results/trends of the study, i.e.” telling the story”. Some tips:
a) It is generally admitted (and several previous studies support this) that inoculation with PGPR can promote mycorhizal fungi colonization and spores production. As a consequence of dual colonization, plant nutrition could be improved and thus yield promoted. Include, and reorganize paragraphs 1 and 2. But, as nutrition has not been measured here, be cautious and note the limitations of the study.
b) What is the point of paragraph 3, concerning the effect of Zn of DSE? Particularly, “The DSE colonization could simultaneously form with AMF colonization in plant root system [28] to deal with biotic and abiotic factors under optimal soil fertility or even higher Zn doses that can’t affect colonization of this fungal group”…. “The low concentration of Zn provided a non-mutualistic relationship between plants and DSE fungal that may reduce biomass production regardless of inoculation”. It is incomprehensible. From the results it can be seen that higher Zn increased DSE colonization, but the explanation is unclear. On the other hand, it is well known that AMF and DSE can colonize roots simultaneously.
c) The last sentence of the discussion is weird. Better, add a closure from your own results that states main trend or their influence for future research. In any case, not always native species are less efficient, thus re inoculation is not necessarily required. Rephrase.
d) Why Zn fertilization is a sustainable management? (line 264).
e) It seems there was an optimal Zn rate both for AM colonization, chlorphyll and yield. This results deserve a comment in discussion section.
2. Title: I am not sure “effect” is correctly used as a verb here. Better “affect”? I insist: In order to broad the scope of the study I suggest authors delete Brazilian Cerrado conditions from title. Better savannah tropical conditions or something like this?
3. English language requires revision by a English native speaker which is familiar with scientific language. Many sentences need to be checked for grammar. For example (but there are others):
l. 41: largest cereal producer in the world
l. 43: production is still not sufficient
l. 60: The management of the interaction of soil-plant-microorganisms is an alternative strategy that could contribute to soil-plant health and productivity
l. 63: soil parts of several cereal crops and improve/promote synthesis of phytohormones
l. 71-71 needs rephrasing
L. 76: A. brasiliense is involved
l. 81: needs rephrasing. Develop positive correlation with AMF to support AMF root colonization???
L- 83-84. Since these symbionts are considered sensitive indicators and can cause behavioural alterations under different environmental or nutritional conditions. It can be deleted.
L. 87: …research is that
L. 129: components of maize were
L. 130: Inoculation with A. brasilense resulted/produced higher number..
L. 132: as compared with noninoculated treatments.
L. 140: rephrase, Zn rates did not respond (¡!)
L. 201: It has also been reported
L. 202-203: fungus play a role increasing mycorrhiza surface area. It is odd.
L. 234-235: repetitive
L.271: Grain yield and yield components of maize and wheat were
L. 436: root colonization instead rhizospheric colonization
L. 437: Delete similarly (because is not similar…)
4. Please indicate in the legend that figure 1 correspond to maize and figure 4 to wheat.
5. Comments of results shown in figures refers to Zn application doses taken from adjusted ecuations, instead real application rates in the experimental field. This aspect should be clearly stated or rewritten.
6. Data in figure 8 have not been used in the study. I suggest authors delete this figure or move to supplementary material.
7. Background information about A. brasiliense effects on maize and wheat in Cerrado conditions fit better in discussion section and can be used to support results
8. Please indicate the time of root sampling from sowing in section 4.4. Measurements
9. Conclusions: “Root colonization by AMF and DSE in maize-wheat cropping system was positively increased with interaction of residual Zn rates and A. brasilense inoculation via seeds” This is not totally true for all the treatments.
Author Response
Dear Reviewer 1:
The authors did a great effort addressing my previous comments and, I guess, the comments from other reviewers to improve the manuscript. Data are now better presented and text is more complete. However, there are still some issues remained.
R: Our most sincere gratitude to you and the reviewer who took time from his busy schedule to help us making this manuscript a better paper. We hope that we have answered every inquiry to your satisfaction and also hope that you will find this version of publishable quality. We hope that this version has met the expectations of the reviewer.
- Discussion is still rather shallow. It should be improved by highlighting novel results/trends of the study, i.e.” telling the story”. Some tips:
- a) It is generally admitted (and several previous studies support this) that inoculation with PGPR can promote mycorhizal fungi colonization and spores production. As a consequence of dual colonization, plant nutrition could be improved and thus yield promoted. Include, and reorganize paragraphs 1 and 2. But, as nutrition has not been measured here, be cautious and note the limitations of the study.
- b) What is the point of paragraph 3, concerning the effect of Zn of DSE? Particularly, “The DSE colonization could simultaneously form with AMF colonization in plant root system [28] to deal with biotic and abiotic factors under optimal soil fertility or even higher Zn doses that can’t affect colonization of this fungal group”….“The low concentration of Zn provided a non-mutualistic relationship between plants and DSE fungal that may reduce biomass production regardless of inoculation”. It is incomprehensible. From the results it can be seen that higher Zn increased DSE colonization, but the explanation is unclear. On the other hand, it is well known that AMF and DSE can colonize roots simultaneously.
R: We appreciate the point of view of the reviewer. Discussion is improved with addition of new information to support the current results in better way. The information added are highlighted in green color the ease of the reviewer. We hope that we answered every inquiry to your satisfaction and also hope that you will find this version of publishable quality. We hope that this version has met the expectations of the reviewer.
- c) The last sentence of the discussion is weird. Better, add a closure from your own results that states main trend or their influence for future research. In any case, not always native species are less efficient, thus re inoculation is not necessarily required. Rephrase.
R: We agree with reviewer. This sentence is removed as it was not making sense. Thanks for highlighting.
- d) Why Zn fertilization is a sustainable management? (line 264).
R: It has been reported by several authors that zinc fertilization along with inoculation of PGPBs in tropical Cerrado condition (as that of the current study) is a sustainable strategy to improve crop yield and nutritional status. We missed to write inoculation. It was corrected.
- e) It seems there was an optimal Zn rate both for AM colonization, chlorphyll and yield. This results deserve a comment in discussion section.
R: These results are discussed in the discussion section.
- Title: I am not sure “effect” is correctly used as a verb here. Better “affect”? I insist: In order to broad the scope of the study I suggest authors delete Brazilian Cerrado conditions from title. Better savannah tropical conditions or something like this?
R: The authors thanks to the reviewer. The title has been changed as per the first round suggestion of the reviewer, however, we still follow the suggestion of the reviewer and made changes to the title as per suggestion. Hope this version has met the expectations of the reviewer. Thank you
- English language requires revision by a English native speaker which is familiar with scientific language. Many sentences need to be checked for grammar. For example (but there are others):
- 41: largest cereal producer in the world
- 43: production is still not sufficient
- 60: The management of the interaction of soil-plant-microorganisms is an alternative strategy that could contribute to soil-plant health and productivity
- 63: soil parts of several cereal crops and improve/promote synthesis of phytohormones
- 71-71 needs rephrasing
- 76: A. brasiliense is involved
- 81: needs rephrasing. Develop positive correlation with AMF to support AMF root colonization???
L- 83-84. Since these symbionts are considered sensitive indicators and can cause behavioural alterations under different environmental or nutritional conditions. It can be deleted.
- 87: …research is that
- 129: components of maize were
- 130: Inoculation with A. brasilense resulted/produced higher number..
- 132: as compared with noninoculated treatments.
- 140: rephrase, Zn rates did not respond (¡!)
- 201: It has also been reported
- 202-203: fungus play a role increasing mycorrhiza surface area. It is odd.
- 234-235: repetitive
L.271: Grain yield and yield components of maize and wheat were
- 436: root colonization instead rhizospheric colonization
- 437: Delete similarly (because is not similar…)
R: We appreciate and thankful to the reviewer. All raised concerns are addressed as per suggestions. Thanks a lot.
- Please indicate in the legend that figure 1 correspond to maize and figure 4 to wheat.
R: Thanks, indicated as per suggestion.
- Comments of results shown in figures refers to Zn application doses taken from adjusted equations, instead real application rates in the experimental field. This aspect should be clearly stated or rewritten.
R: Thanks, the five Zn rates (0, 2, 4, 6 and 8 kg ha-1) shown in the figures were applied in the field experiment. As is usual in nutrient rate research (quantitative factor), when there is a significant effect for a nutrient's rates, we should choose the best fit based on the significance of the results. Once the quadratic function has been adjusted, we must derive the equation, then set it to zero and calculate the "x" of the equation. This "x" calculated is equivalent to the optimal rate estimated to obtain the highest corresponding value of the evaluation to this mathematical adjustment.
- Data in figure 8 have not been used in the study. I suggest authors delete this figure or move to supplementary material.
R: Thanks, moved to supplementary material as per reviewer’s suggestion.
- Background information about A. brasilienseeffects on maize and wheat in Cerrado conditions fit better in discussion section and can be used to support results
R: Thanks. Discussion is improved and supported by new literature.
- Please indicate the time of root sampling from sowing in section 4.4. Measurements
R: Thanks, briefly indicated as per suggestion.
- Conclusions: “Root colonization by AMF and DSE in maize-wheat cropping system was positively increased with interaction of residual Zn rates and A. brasilense inoculation via seeds” This is not totally true for all the treatments.
R: We agree with reviewer. The conclusions are changed as per suggestion.

Reviewer 2 Report (New Reviewer)
In the present status, the paper is far from publication in this journal. The deficiency at least (but not limit) as below:
Although the work is novel and important, I suggest improving the presentation of the objective/niche.
The most glaring problem with the manuscript is that the discussion and conclusions are not fully supported by the results of the study.
Some parts of the materials and method section should be more detailed (for example characterize the regional problem, bacterial culture media, why do they use that source of Zn and in those doses?, why was soil sampling not done on wheat? This information would be very useful when evaluating Zn residuality.
I think it would convenient to estimate bacterial viability to Zn.
Figure 1E and supplementary material are not mentioned in the text.
Improve the presentation order of figures and check the scale values.
I suggest reconsidering a properly rewritten manuscript, as the language is not fluent, and there are typographic mistake. Numerous figures are mentioned in the discussion that do not correspond to the text or to what was observed.
The discussion section should be further improved. For example, it should be properly developed by arguing findings and comparing with what is already known. In addition, a clearer conclusion should be included based on the discussion.
Author Response
Dear Reviewer 2,
Although the work is novel and important, I suggest improving the presentation of the objective/niche.
The most glaring problem with the manuscript is that the discussion and conclusions are not fully supported by the results of the study.
R: Thanks, the discussion and conclusions are improved. We hope that we answered every inquiry to your satisfaction and also hope that you will find this version of publishable quality. We hope that this version has met the expectations of the reviewer.
Some parts of the materials and method section should be more detailed (for example characterize the regional problem, bacterial culture media, why do they use that source of Zn and in those doses?, why was soil sampling not done on wheat? This information would be very useful when evaluating Zn residuality.
R: Thanks, the problem of regional problem is already explained in the introduction section, the tropical areas are being spotted with zinc deficiency. We have added the data about most probably number (MPN) of Azospirillum sp., this Zn sources is one the most commonly used, easily available in the markets, and not toxic in the doses applied in the current experiment. We did not do these analysis at that time and now the soil source is not available to do residual Zn analysis. We have been cited some of the literature in our discussion and introduction, where the findings of other researcher support our results.
I think it would convenient to estimate bacterial viability to Zn.
R: We appreciate and thankful to the reviewer. However, this research has carried out in 2013/2014 cropping seasons. We really apologize, we don’t samples to do bacterial estimation now.
Figure 1E and supplementary material are not mentioned in the text.
R: Thanks, it has mentioned for the consideration of the reviewer.
Improve the presentation order of figures and check the scale values.
R: Thanks, figures resolution was improved and legends were updated.
I suggest reconsidering a properly rewritten manuscript, as the language is not fluent, and there are typographic mistake. Numerous figures are mentioned in the discussion that do not correspond to the text or to what was observed.
The discussion section should be further improved. For example, it should be properly developed by arguing findings and comparing with what is already known. In addition, a clearer conclusion should be included based on the discussion.
R: We appreciate the point of view of the reviewer. All the raised concerns are addressed. The manuscript is properly revised. Proper attention has given to discussion, which is improved with recent literature to support the current results in better way. The information added are highlighted in green color the ease of the reviewer. We hope that we answered every inquiry to your satisfaction and also hope that you will find this version of publishable quality. We hope that this version has met the expectations of the reviewer.

Reviewer 3 Report (New Reviewer)
The work submitted for review concerns important aspects from the point of view of phytopathology and agronomics. In my opinion, the experiments are well planned and well done and the results are interesting. The manuscript is well written and its sections are consistent. I do not see any serious factual and editorial errors at work.
Author Response
Dear Reviewer 3,
The work submitted for review concerns important aspects from the point of view of phytopathology and agronomics. In my opinion, the experiments are well planned and well done and the results are interesting. The manuscript is well written and its sections are consistent. I do not see any serious factual and editorial errors at work.
R: Our most sincere gratitude to you and the reviewer who took time from his busy schedule to review out paper. We really appreciate your positive concern about our study. Thanks.

Reviewer 4 Report (New Reviewer)
Dear Authors,
The manuscript describes interesting results about the use of biofertilization with A. brasilense combined with different rates of Zn fertilization. Some questions appeared in the revision of the manuscript
-Figure 1b and 2b is difficult to understand that a linear relationship may exist despite the fact that the authors have forced the use of this line with relatively low r2
-In all the parameters analyzed, the treatment with A. brasilense and 6 kg/ha of Zn seems to show a discordant behavior with the rest of the treatments. What reason could this be due to?
-Because the discussion focuses on the production of IAA in the plant but there is no analysis of the production in bacteria or other PGP mechanisms of the bacteria. In this aspect it may be interesting to know and analyze the production of siderophores
-Line 405: lenght of segments
-It should be recommended to analyze other macro and micronutrients
Author Response
Dear Reviewer 4,
The manuscript describes interesting results about the use of biofertilization with A. brasilense combined with different rates of Zn fertilization. Some questions appeared in the revision of the manuscript
R: Thanks! The authors tried to address all the respective concern of the reviewer. Hope this version met the expectations of the reviewer.
-Figure 1b and 2b is difficult to understand that a linear relationship may exist despite the fact that the authors have forced the use of this line with relatively low r2
R: These figures are being made after statistical analysis. The linear relationship exists in case of the evaluations in the Figure 1A and Figure 2B were adjusted at 1% of significance that is, even if the adjustment (R2) was not high, we chose to take this into account to explain the results. If the significance was adjusted to 5%, then we will surely agree with you and not take into the account this explanation.
-In all the parameters analyzed, the treatment with A. brasilense and 6 kg/ha of Zn seems to show a discordant behavior with the rest of the treatments. What reason could this be due to?
R: The possible explanation for this might be the toxic effect of Zn at high level. We don’t have the insight analysis of these parameters to explain in better way. It is well known that Zn application in higher doses cause toxicity that can have harmful impact on growth and productivity of crop plants.
-Because the discussion focuses on the production of IAA in the plant but there is no analysis of the production in bacteria or other PGP mechanisms of the bacteria. In this aspect it may be interesting to know and analyze the production of siderophores
R: We appreciate the point of view of the reviewer. However, the discussion is revised with more relevant information about the current study. In addition, at this stage we don’t have the sample to do suggested these kind of analysis.
-It should be recommended to analyze other macro and micronutrients
R: We would happily follow your suggestion however, at this stage we don’t have the sample to do suggested analysis. Hope the reviewer should understand our concern.
Thanks!

Reviewer 5 Report (New Reviewer)
The manuscript by Silva et al outlines experiments to evaluate how wheat and maize cultivated under Zn limitation are affected by Azospirillum brasilense.
The study is well designed and executed, and the results of interest. In contrast to the rest of the paper, the introduction is week and does not lead me to see why this work was conducted. This should not be too hard to remedy, but will require an extensive re-write. There are many minor grammatical and word choice errors throughout the manuscript that should be attended to.
The last paragraph of the introduction does not contain a stated hypothesis, only a question. The question from line 86 should just be rephrased. Importantly, the authors should present a series of established findings and areas of unknown to lead to the hypothesis. As examples, what is the reason that boosting of mycorrhiza could contribute to zinc deficiency? What are DSE and what do they do differently to AMF? The sentence from lines 47 – 50 makes little sense. What is “short life cycle” in this context, and what does C sequestration have to do with the topic under study? There is mention of spores in several places, but it is not stated what taxa these spores belong to – AMF, DSE, soil bacteria?
Did the authors determine occurrence of Azospirillum in soil / rhizosphere at any stage after treating seeds? If these data are not available, the authors are asked to include softening statements on the role of Azospirillum. It is not impossible that the effects observed were due to chemicals in the treatment rather than due to the bacterial inoculant.
Specific comments:
11. Line 2: “affecting” rather than “effecting”
22. Line 16: you give a solution here when this part of the abstract is about leading to the hypothesis.
33. Line 45: Change “interrupted” to a word such as “impacted”.
44. Line 55: “….synthesis” of what organisms – Azospirillum, AF, DSE or plant?
55. Line 90: Spores of which taxa?
66. Figure 1 A, C and E – no X axis title.
77. Figure 1 legend – of what crop – maize?
88. Line 111 and elsewhere – is this difference at 5 or 95% probability?
99. Figure 2: X axis legends for B and D
110. Figure 4 A, C and E – no X axis title.
111. Figure 4 legend – of what crop – wheat?
112. Line 186: “cob” or spike?
113. Line 200 “they” – should this word be removed?
114. Line 403: As stated treatment with acid and base simultaneously appears odd. Was one before the other?
Author Response
Dear Reviewer 5,
he manuscript by Silva et al outlines experiments to evaluate how wheat and maize cultivated under Zn limitation are affected by Azospirillum brasilense.
The study is well designed and executed, and the results of interest. In contrast to the rest of the paper, the introduction is week and does not lead me to see why this work was conducted. This should not be too hard to remedy, but will require an extensive re-write. There are many minor grammatical and word choice errors throughout the manuscript that should be attended to.
R: The introduction is improved by inserting recent citations and more relevant information. The authors tried their level best to address each and every minor mistake.
The last paragraph of the introduction does not contain a stated hypothesis, only a question. The question from line 86 should just be rephrased. Importantly, the authors should present a series of established findings and areas of unknown to lead to the hypothesis. As examples, what is the reason that boosting of mycorrhiza could contribute to zinc deficiency? What are DSE and what do they do differently to AMF? The sentence from lines 47 – 50 makes little sense. What is “short life cycle” in this context, and what does C sequestration have to do with the topic under study? There is mention of spores in several places, but it is not stated what taxa these spores belong to – AMF, DSE, soil bacteria?
R: The authors appreciate the reviewer’s concern about our study. Hypothesis is revised. The area of the current is relative deficient in Zn content, while growing cereals like maize and wheat are exhaustive crops in case of Zn fertilization that should further decrease Zn assimilation to the edible tissue of each crop (which is not presented in the current study). These fungal strains are well studied in agriculture for improving plant nutritional status. As we had more information about AMF and its role in crop production system, while we tested DSE in comparison, which has almost the similar characteristics. The rest of the results of both fungal strains are described in the paper. Cereals, especially maize is considered a short life cycle crop due to its two major cultivation in Cerrado while some of the Brazilian regions can obtain more than three harvest per year. Brazilian cropping system only collect grain and remain the residues of the wheat and maize on the field that could incorporate and help in soil fertility. Carbon sequestration is mentioned here as we have determined C-CO2 in the current study. The study has only determined AMF spores and know make it clarified. Hope this version has met the expectations of the reviewer.
Moreover, use of AMF is one of the promising technology in the soil plant atmosphere continuum and responded to crop plants to improve water uptake as a result of root expansion with fungal hyphae, protecting against biotic and abiotic stresses and enhancement of plant antioxidant status. AMF establish symbiotic associations with more than 80% of the terrestrial plants. Its colonization rate depends upon the fungi, plant species and soil characteristics. It can increase accumulation of macronutrients (N, P) and micronutrients (Zn, S, Cu, Fe and Mn) as Zn is the nutrient of the question.
Did the authors determine occurrence of Azospirillum in soil / rhizosphere at any stage after treating seeds? If these data are not available, the authors are asked to include softening statements on the role of Azospirillum. It is not impossible that the effects observed were due to chemicals in the treatment rather than due to the bacterial inoculant.
R: No, we did not determine occurrence of Azospirillum in soil / rhizosphere at any stage after treating seeds. However, the authors include most probably number (MPN) of Azospirillum sp. in Table 1. The results had clearly highlighted the effect of inoculation and Zn rates in the manuscript. To be more clear, we had interpreted the data of each evaluation in individual figure.
Specific comments:
- Line 2: “affecting” rather than “effecting”
- Line 16: you give a solution here when this part of the abstract is about leading to the hypothesis.
- Line 45: Change “interrupted” to a word such as “impacted”.
- Line 55: “….synthesis” of what organisms – Azospirillum, AF, DSE or plant?
- Line 90: Spores of which taxa?
- Figure 1 A, C and E – no X axis title.
- Figure 1 legend – of what crop – maize?
- Line 111 and elsewhere – is this difference at 5 or 95% probability?
- Figure 2: X axis legends for B and D
- Figure 4 A, C and E – no X axis title.
- Figure 4 legend – of what crop – wheat?
- Line 186: “cob” or spike?
- Line 200 “they” – should this word be removed?
- Line 403: As stated treatment with acid and base simultaneously appears odd. Was one before the other?
R: The addressed all suggestions of the reviewer to the best of their efforts and knowledge. Hope this version has met the expectations of the reviewer.
Thanks!

Round 2
Reviewer 5 Report (New Reviewer)
Thank you for your attention to the points I raised in the first review.
This manuscript is a resubmission of an earlier submission. The following is a list of the peer review reports and author responses from that submission.
Round 1
Reviewer 1 Report
The Introduction and Abstract contain sentences which are difficult to understand and/or grammatically incorrect or no specific enough.
In the Abstract, e.g. authors want to know the result, but I do not understand the result from the sentence in lanes 28/9.
In the Introdcution:
. What means “low beyond consumption”, “functional of soil existing microorganisms” “..tryptophan as precursor of Zn”, or “residual Zn rates”
There should be a “,” before "therefore" at several places.
Also in l. 40 instead of interrupted, maybe restricted.
The authors also want to learn in the Introduction why the bacterium and these fungi were used for the study. But there is no information.
Results
l. 83 “root”
l. 91: species name italics
l. 93/ “inoculated treatment”. Do the authors mean “inoculation”?
Figure 1 is difficult to understand, what means “function of single effect of Zn rates” in the legend?
Discussion
l.136 Where are the IAA data?
There is also much discussion about root colonisation, but only suppl. data are shown. This needs a rigorous investigation.
l. 166 ff discusses the mode of interaction, but the paper does not show any investigation.
Overall, this is a collection of data and I do not understand the scientific conclusion from the data. Even the title only mentions that microorganisms and Zinc affect crops but how? The Introduction is not very helpful in addressing a scientific question and introducing the system. The Discussion includes numerous aspects of the symbionts and Zn without clear relationship to the results. Writing also requires revision.
Therefore, I suggest to reject the manuscript.
Reviewer 2 Report
Review: Azospirillum brasilense and zinc rates effecting fungal root colonization and yield of cereal crops in succession under Brazilian Cerrado conditions
This study reports the effect of Zn application and Azospirillum inoculation on arbuscular mycorrhizal fungi and Dark septate endophytes colonization and on grain yield in maize and corn under agricultural field conditions.
As positive, the introduction, experimental design and methods description are accurate and scientifically sound. However, as negative, determined data/parameters seem insufficient and consequently, the discussion is rather weak or not totally supported by data.
Please find here specific comments that I hope will be helpful to improve the manuscript:
1. In order to broad the scope of the study I suggest authors delete Brazilian Cerrado conditions from title. It is interesting, however, maintain the concerning information in other sections.
2. English language requires revision by an English native speaker. For example, lines 61-68 are difficult to follow. Do you mean PGPB can alter symbiosis between plants and fungi? Do you mean AMF and DSE can increase implications of AMF?? Or, it is not clear what authors mean with the sentence Zn management and biotechnology. In lines 135-136 it is unclear that bacteria promoted nutrient acquisition with improved rhizospheric environment. Signal AMF colonization ? (line 147). Paragraph in lines 148-150 needs revision as well.
3. Hypothesis should be rewritten: the effect of Zn and inoculation on soil microbiota and AMF/DSE populations has not been studied here, but just root colonization/spores, i.e. the study lacks a taxonomic/diversity methodology for microbial populations.
4. I miss some information for Figure legends corresponding graphs in the right (columns). Do they represent the total number of spores, grain yield, etc, for the Zn treatments as a whole?
5. I cannot see the supplementary material, so I am not able to review this information.
6. Readers would appreciate to see results of ANOVA statistic i.e. differences among treatments and, importantly, their interaction, which is a main objective of the study. Correlations statistics (Pearson, Spearman, etc) would be also interesting.
7. Discussion needs improvement; many sentences are speculative. For example:
a) Lines 135-136 state that results indicate that Zn + bacteria produce IAA and enhanced plant growth, but these parameters have not been measured.
b) Was root colonization from seedling to maturity determined here? Lines 152-153. Maybe these data are in table S1 which I cannot see?
c) Lines 160-164: It is not demonstrated that low Zn lead to non-mutualistic relationship between plant and DSE. The concepts mutualistic, symbiotic have to be revised throughout the text.
d) Argument concerning the influence of Azospirillum on grain yield based on plant nutrition is weak, because the study lacks metrics such as nutritional status of plants, plant growth, etc.
8. It is weird finishing conclusion with a reference from literature. Better, add a closure that states main results and their influence for future research.
Reviewer 3 Report
This paper examines the dual effects of differing levels of zinc application and Azospirillum brasilense inoculation on colonisation of maize roots by dark septate endophyte and arbuscular mycorrhizal fungi. The effects of these treatments on yield were also measure. The effects of the residual zinc treatment on colonisation of wheats roots by dark septate endophyte and arbuscular mycorrhizal fungi was examined by planting wheat into the same plots as the previously grown maize plants. The effect inoculation or otherwise of wheat by Azospirillum brasilense was also examined in this second planting.
The results show that there is an increase in AM fungal colonisation with A. brasilense inoculation in maize, and that AM colonisation starts to decrease with higher zinc levels. With DSE there is less marked difference between inoculated and un-inoculated maize plants. The ‘linear’ increase in DSE colonisation in treated plants is somewhat dependent on the measurement of the 8 kg per ha Zn treatment. It would be useful for the authors to speculate/discuss the difference between the 6 kg per ha Zn and this 8 kg per ha value.
With respect to wheat, it would have been advantageous to measure the residual Zn levels after the treatment and growth of maize. Without these measurements the comparisons of the two crops and zinc treatment cannot be clearly made. It would be good for the authors to further discuss the differences between maize and wheat root fungal colonisation, from their results wheat is more colonised by DSE than AM, whereas the opposite is seen for maize. Is this consistent with the literature of these plants grown in other soils?
For both plants, example images of the clarified and stained root samples, both those containing fungi and those that were negative, would add value to the manuscript.
Some idea of the size of the soil samples take for measurements of roots, spores and (in grams or volume) would help the reader further understand the results presented. Are they the same soil core type cup auger (0.10 m × 0.40 m - cup diameter and length respectively) as used in the initial soil analysis?
The measurements of CO2-C and spore counts for the two plants could be plotted on the same graphs to enable comparisons of how much they have changed between the two plants.
In the discussion the authors talk about the “hormone promoted root system” with respect to Zn and bacteria producing IAA. Do the authors have any data from their experiment to validate changes in the root system (amount of root per soil sample? Images of roots from harvested plants?).
Overall, the study shows interesting results of the interactions between Zinc treatments, A. brasilense inoculation and two different crops in a well characterised soil system in Brazil. With some minor updates the study should be publishable in plants.